# Improving Training Efficiency via Language Specific Model Merging

## Abstract

Fine-tuning a task-specific multilingual large language model (LLM) involves training the model on a multilingual dataset with examples in all the required languages. Updating one or more supported languages with additional data or adding support for a new language involves retraining the model, which can be computationally inefficient. Recent research on merging multiple task-specific models has shown promise in terms of both computational efficiency and improved performance. These approaches only consider multiple tasks in a single language, but their effectiveness in merging language-specific models trained on a single task is underexplored. In this work, we explore existing model merging approaches in a multilingual setting for three independent tasks. Our experiments show that model merging approaches achieve performance on par with models trained on a combined dataset of multiple languages, as well as the language-specific fine-tuned models. Our analysis indicates that training efficiency improves by reducing the training time of adding or updating new languages by 2.5 times and reducing training costs by 3 times.

## 1 Introduction

Large Language Models (LLMs) have gained significant attention in many NLP applications. Although these models have shown exceptional zero-shot and few-shot capabilities in tasks like text classification, summarization, and reasoning, among others, their performance can be enhanced further by fine-tuning them with task-specific datasets. In particular, parameter-efficient fine-tuning (PEFT) approaches such as LoRA (Hu et al., 2021) are frequently used, as they drastically reduce the number of trainable parameters, resulting in faster training and reduced memory usage. The adapter obtained via LoRA fine-tuning can be merged with the base model to get the final fine-tuned model. To ensure multilingual support when fine-tuning models for a specific task, two approaches can be used: (1) fine-tuning a language-specific LoRA adapter, creating multiple language-specific models for each task, (2) fine-tuning a single multilingual adapter with a multilingual task-specific dataset.

Both these approaches have some limitations. Individual language-specific models give the best performance for the specific language, however, using multiple language-specific models during inference can be resource-intensive and computationally expensive. Whereas, multilingual model training may require high-quality task-specific training data, as well as higher computational resources. Furthermore, adding and/or updating language support in a multilingual model requires model retraining, which is an expensive process. Recently, model merging has been explored for multitask models. This involves fine-tuning task-specific models with high-quality training data and merging the individual models to obtain a single multitask model. Several approaches to merge multiple task-specific models have been proposed recently (Ortiz-Jimenez et al., 2023; Yadav et al., 2023; Yu et al., 2024). These model merging techniques have not only improved the task-specific performance (Yadav et al., 2023), but they are also shown to be resource efficient.

While model merging has been extensively applied in a multitask setting, multilingual model merging is underexplored. Few works have considered multilingual model merging (Parović et al., 2024; Tao et al., 2024; Zhao et al., 2025); however, these works extensively consider transfer learning to improve the performance of a task from a source language to a target language. These techniques are often not directly applicable in an enterprise application because we do not have a priori knowl-

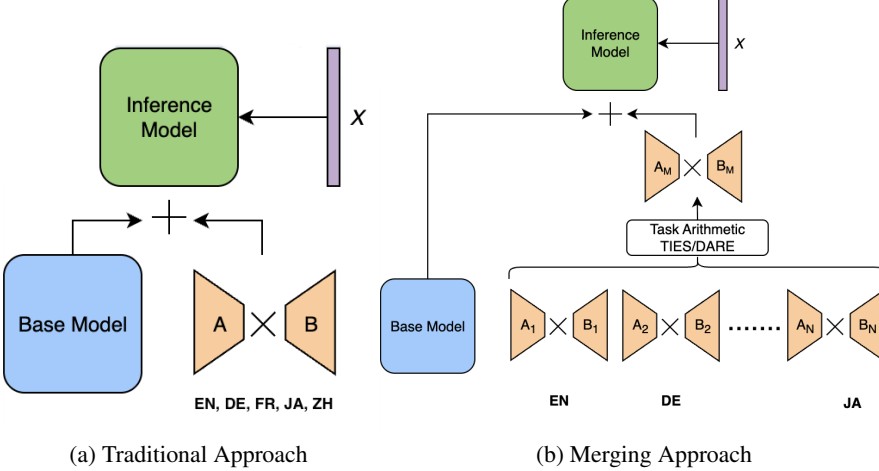

(a) Traditional Approach        (b) Merging Approach

Figure 1: Comparison of multilingual model fine-tuning and Language-specific model merging. (a) shows the traditional approach, where a multilingual adapter is trained with a combined dataset and further merged with the base model. (b) shows the language-specific merging where language-specific adapters are first trained and merged to get a single multilingual adapter, which is then merged with the base model

edge of the source language. Moreover, when multiple geographies are involved, we frequently deal with texts that contain more than one language making automated language detection based routing vague and unreliable. Hence, we need a single multilingual model capable of supporting all required languages. In this work, we leverage existing model merging techniques in a multilingual setting to create such a multilingual model (Figure 1). Specifically, we experiment with three merging techniques, TIES (Yadav et al., 2023), DARE (Yu et al., 2024), and KnOTS (Stoica et al., 2024). To understand their generalizability, we consider three different tasks: Sentiment Analysis, Abstractive Summarization, and Commonsense Reasoning. Through extensive experiments and several ablations, we seek to answer the following questions: (1) How does the performance of the merged model compare to language-specific models and a single multilingual model? (2) Do independent language updates improve the performance on the specific language and translate to other languages in the merged model? Does this improve training efficiency? (3) How does language cluster-based model merging influence the performance quality of each language?

## 2  RELATED WORK

**Multilingual Fine-tuning:** Expanding language support for any task involves fine-tuning a multilingual base model on task-specific multilingual data (Eisenschlos et al., 2019; Ladhak et al., 2020; Choenni et al., 2023; Muennighoff et al., 2023; Indurthi et al., 2024). Some works propose strategies to select the most relevant examples to maximize performance for a task or improve model generalization across languages. Muennighoff et al. (2023) show that fine-tuning multilingual language models on English instruction datasets improves their performance in zero-shot settings, and generalizes on unseen languages. Choenni et al. (2023) propose using only the training examples most influential for test predictions. They further show that during fine-tuning, the training samples across languages enhances knowledge sharing. In the same vein, our experiments aim to show that, similar to fine-tuning on a combined language dataset, merging individually fine-tuned models on different languages also acquires task–specific knowledge from different languages.

**Model Merging:** With the increasing language model size, model merging has gained popularity to improve multitask model performance and model generalization (Wortsman et al., 2022; Matena & Raffel, 2022; Choshen et al., 2022; Ilharco et al., 2023; Yadav et al., 2023; Ortiz-Jimenez et al., 2023; Tang et al., 2024; Jin et al., 2025; Ram et al., 2024; Yu et al., 2024; Stoica et al., 2024). Some initial approaches involve averaging the model weights with techniques like standard averaging (Choshen et al., 2022), Fisher-weight averaging (Matena & Raffel, 2022), and RegMean (Jin et al., 2025).

Further enhancements to model merging were introduced by Ilharco et al. (2023), who proposed Task Arithmetic, a merging technique using arithmetic operations like addition and subtraction, to add and remove tasks, respectively. They observed improved performance on individual tasks with the merged model. Other approaches like TIES (Yadav et al., 2023) and DARE (Yu et al., 2024) prune the individual fine-tuned models to improve merged model efficiency. However, with merging PEFT fine-tuned models, task alignment issues were introduced. Hence, some works introduced techniques to improve the task alignment (Tang et al., 2024; Stoica et al., 2024). Apart from these merging techniques, other works introduced dynamically selecting relevant task-specific adapters at inference time, where a router layer is further trained to select the relevant adapter (Pfeiffer et al., 2021; Feng et al., 2024; Buehler & Buehler, 2024; Li et al., 2024).

**Crosslingual Merging and Transfer:** Most of the recent model merging techniques have been used for multiple tasks. However, there has been limited research in cross-lingual model merging. Some earlier works have used transfer learning for a specific task, where the language-specific or multilingual adapters are trained on two languages, and a task-specific adapter is trained on top of one of the languages, with data for that specific language. The learnings are then transferred to other languages by changing the language adapters (Pfeiffer et al., 2020; Parović et al., 2022; 2023). Parović et al. (2024) proposed using Task Arithmetic to transfer task-specific knowledge from a source language to a target language. While Tao et al. (2024) use continual training with language-specific data, followed by task-specific English SFT training and further model merging with TIES or weight averaging to obtain a task-specific model in a target language. Apart from these works, other approaches like AdaMergeX (Zhao et al., 2025) that perform cross-lingual transfer learning by leveraging model merging have been proposed. Unlike these works, which use transfer learning, our work aims to determine the effectiveness of model merging techniques in a multilingual setting to solve a specific task, while analyzing their computational efficiency.

## 3 APPROACH

### 3.1 PRELIMINARIES

In this section, we give an overview of the merging techniques used in our experiments, namely TIES, DARE, and KnOTS.

**TIES:** Yadav et al. (2023) proposed Trim, Elect Sign, and Merge (TIES), a three step approach for merging models fine-tuned on multiple tasks. The first step retains top-k percent of the model weights for each of the fine-tuned models to be merged. A sign is elected in the second step, where the summation of all the positive signed values for the specific weight position is computed. Similarly the summation for all the negative weight values is computed for the same weight position. The sign with the maximum magnitude is selected as the final sign for the weight position under consideration. A disjoint merge is then performed as the last step, where the mean of the selected sign values is computed to get the merged weight value.

**DARE:** Drop And REscale (DARE) (Yu et al., 2024) is an approach introduced to prune the redundant weights from the fine-tuned models. This simple approach first randomly sets certain weight values to 0, determined by a drop-rate p, unlike TIES, where the lowest weight values are dropped. This adds sparsity to the model weights. The remaining weights are further scaled by a factor of $p/(1-p)$. The pruned fine-tuned models can then be merged using any existing merging techniques. It is considered a plug-and-play module for existing merging techniques, with minimal performance loss.

**KnOTS:** Stoica et al. (2024) proposed Knowledge Orientation Through SVD (KnOTS), a precursor to model merging. They observe that transforming the weights of the fine-tuned models to a common space leads to better alignment during model merging. This approach makes use of SVD for this transformation. SVD better aligns the representations between different LoRA adapters. The approach works by first concatenating the individual fine-tuned model weights layer by layer and then applying SVD over it to obtain a set of task-specific concatenated matrices. These matrices are then merged using an existing merging technique.

### 3.2 EXPERIMENTAL SETUP

#### 3.2.1 DATASETS

To evaluate the effectiveness and generalization of multilingual model merging, we considered three tasks: sentiment analysis, abstractive summarization, and commonsense reasoning. For each of the three tasks, we experimented with five languages: English (EN), German (DE), French (FR), Japanese (JA), and Chinese (ZH). We used the **MultilingualSentiment** (clapAI, 2024) dataset for sentiment analysis, **mCSQA** (Sakai et al., 2024) for reasoning and **WikiLingua** (Ladhak et al., 2020) dataset for summarization. We use 5,000 training, 500 validation, and 500 test examples for sentiment analysis and reasoning tasks, whereas, for summarization, 3,000 training, and 500 validation and test examples are used.

#### 3.2.2 BASELINES

For each task and language, we use two baselines: a model fine-tuned with a combined task-specific dataset of all languages, and an individual model trained on a task-specific, language-specific dataset. The combined dataset for a task includes the examples from the five individual language datasets. We use Llama-3.1-8b-Instruct (Grattafiori et al., 2024) as the base model for all tasks and languages. Each model was fine-tuned using LoRA (Hu et al., 2021) with r=64 and alpha=64. A learning rate of 2e-5 was used with a training batch size of 8 and a maximum sequence length of 8196.

#### 3.2.3 MERGING

We experimented with several combinations of the three merging techniques mentioned in Section 3.1. More specifically, we used the following combinations: TIES, TIES-KnOTS, DARE-TIES, and DARE-TIES-KnOTS. Since DARE and KnOTS are preliminary tasks to other merging techniques, they cannot be used as a standalone merging technique. Two hyperparameters are used: weights, which are a vector that determines the amount of weight to be given to each fine-tuned model, and density, which determines the percentage of weights' values to be pruned. For each of these combinations, we use two sets of hyperparameters: (weights=1, density=1) and (weights=1, density=0.5). This resulted in 8 merged models for each task.

#### 3.2.4 METRICS

We compute the macro-average F1-score, Precision and Recall for the sentiment analysis task to account for class imbalance. The commonsense reasoning task is evaluated using multi-class accuracy, and summarization is evaluated using ROUGE-1, ROUGE-L, and Bert-Score.

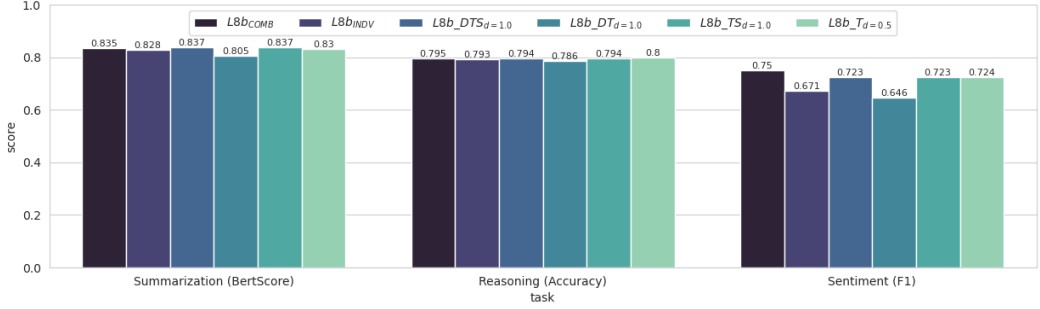

Figure 2: Overall performance of the baselines and the merged models on the summarization, commonsense reasoning, and sentiment analysis tasks. For the three tasks, we plot the BertScore, Accuracy, and F1 scores, respectively.

Table 1: The language-specific metric scores for each task. We show the baselines along with the best merged model for each merging technique (D=DARE, T=TIES, S=KnOTS). For summarization, we report the mean BertScore, while for reasoning and sentiment analysis, we report the Accuracy and F1 scores, respectively.

| Model | Summarization (BertScore) | | | | | Reasoning (Accuracy) | | | | | Sentiment (F1) | | | | |
|---|---|---|---|---|---|---|---|---|---|---|---|---|---|---|---|
| | EN | DE | FR | JA | ZH | EN | DE | FR | JA | ZH | EN | DE | FR | JA | ZH |
| $L8b_{COMB}$ | 0.839 | **0.834** | **0.837** | 0.830 | 0.835 | 0.896 | **0.840** | 0.754 | 0.754 | 0.732 | 0.759 | **0.791** | 0.756 | 0.768 | 0.675 |
| $L8b_{INDV}$ | 0.837 | 0.817 | 0.835 | 0.814 | 0.836 | 0.876 | 0.836 | **0.770** | 0.758 | 0.724 | **0.791** | 0.755 | 0.525 | 0.775 | 0.509 |
| $L8b\_DTS_{d=1.0}$ | **0.840** | 0.833 | 0.836 | **0.836** | **0.838** | 0.898 | 0.824 | 0.756 | 0.758 | **0.736** | 0.641 | 0.773 | 0.762 | 0.758 | **0.682** |
| $L8b\_DT_{d=1.0}$ | 0.811 | 0.792 | 0.797 | 0.811 | 0.813 | 0.874 | 0.822 | 0.752 | **0.776** | 0.708 | 0.470 | 0.747 | 0.769 | 0.769 | 0.474 |
| $L8b\_TS_{d=1.0}$ | **0.840** | 0.833 | 0.836 | **0.836** | **0.838** | 0.898 | 0.824 | 0.756 | 0.758 | **0.736** | 0.641 | 0.773 | 0.762 | 0.758 | **0.682** |
| $L8b\_T_{d=0.5}$ | 0.836 | 0.824 | 0.828 | 0.830 | 0.832 | **0.908** | 0.832 | 0.760 | 0.774 | 0.724 | 0.651 | 0.756 | **0.774** | **0.778** | 0.659 |

## 4 RESULTS AND DISCUSSION

For the summarization task, we see that the overall performance of the merged models is comparable to both the baselines, as seen in Figure 2. Among the merged models, Llama-8b merged with TIES-KnOTS(TS) and DARE-TIES-KnOTS(DTS) have the best performance, followed by TIES(T) and DARE-TIES(DT). Across the languages, the merged models outperformed the baselines on English, Japanese, and Chinese with BertScore improving between 0.1 to 0.6%, as seen in Table 1. While the BertScores of the merged models were on par with the baselines, we see that merging can improve the ROUGE scores for some of the languages, as indicated in Appendix A.1

For the Commonsense reasoning task, we see that the overall accuracy of the merged models is comparable to that of the baseline models. Similar to summarization, the baselines showed a slightly better accuracy for German and French, while for English, Japanese, and Chinese, the merged models slightly outperformed the baselines. The difference in accuracy between the baselines and the best merged models ranges from 0.4 to 2.2% absolute difference. The TIES model with a density of 0.5 outperforms the multilingual baseline by 0.5%.

Unlike the other two tasks, for sentiment analysis, the model trained on combined multilingual data achieves the best performance, while some of the merged models outperform the language-specific model. Considering the language-specific performance, the merged model performance for French, Japanese, and Chinese is slightly higher than both the individual and the multilingual models. However, there is a 1.7% absolute difference between the multilingual baseline and the best merged model for German, while for English, there is a significant performance difference, of the magnitude of 15% between the best merged model and the best baseline. The overall lower performance of the merged models can be attributed to the lower performance of these models on English. Furthermore, the effect of merging can also be task-specific; in tasks with limited label space, the influence of merging is not as effective, while for tasks like summarization and reasoning, where the label set is varied, merging may have more influence on the performance.

From Table 1 we see that the models merged with DARE-TIES and TIES with density 1 have similar metric scores across all tasks. Since none of the weights are pruned with density 1, and due to the later merging steps being similar for both DARE and TIES, we see a similar performance for these two approaches. We further compare the efficiency of model merging with the multilingual model fine-tuned on the summarization task. As shown in Figure 3, we see that fine-tuning individual language models is time-efficient compared to fine-tuning the model on a combined dataset of all languages, as it enables parallel training. We also see that updating an individual language model and merging it with the other language-specific models is computationally more efficient compared to retraining a single multilingual model on the updated dataset. This suggests that, multilingual model merging may achieve on-par performance to fine-tuning a model on a combined dataset of all the languages, but can be computationally efficient, especially when a new language is to be added to an existing model or a language-specific model is to be updated.

## 5 ABLATIONS

To further understand the advantages and disadvantages of language-specific model merging, we conduct additional ablations on the tasks. More specifically, we try to understand the impact of

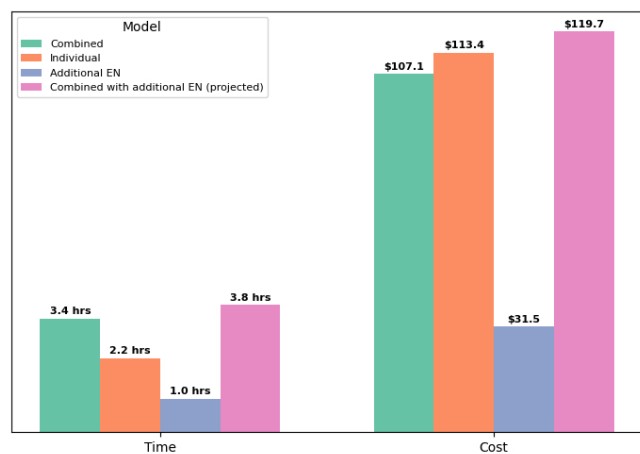

Figure 3: Comparison of Training Speed and Cost with fine-tuning LLMs on language-specific data for the abstractive summarization task.

language grouping on model merging, language-specific updates, and the performance difference of merging smaller LLMs. We also merge task-specific language-specific models to evaluate how the performance is affected across the tasks and languages.

Table 2: Results for the merged model with updated EN adapter and the language cluster-based merged models

| Model | Summarization | | | | | Reasoning | | | | | Sentiment | | | | |
|---|---|---|---|---|---|---|---|---|---|---|---|---|---|---|---|
| | EN | DE | FR | JA | ZH | EN | DE | FR | JA | ZH | EN | DE | FR | JA | ZH |
| $MERGED_{Best}$ | 0.840 | 0.833 | 0.836 | 0.836 | 0.838 | 0.908 | 0.832 | 0.760 | 0.774 | 0.724 | 0.651 | 0.756 | 0.774 | 0.778 | 0.659 |
| $TIES_{EN-Updated}$ | - | - | - | - | - | - | - | - | - | - | 0.684 | 0.781 | 0.771 | 0.782 | 0.666 |
| $TIES_{EN-DE-FR}$ | 0.840 | 0.834 | 0.835 | - | - | 0.900 | 0.838 | 0.770 | - | - | 0.651 | 0.738 | 0.746 | - | - |
| $TIES_{JA-ZH}$ | - | - | - | 0.836 | 0.838 | - | - | - | 0.740 | 0.732 | - | - | - | 0.771 | 0.667 |

## 5.1 LANGUAGE SPECIFIC UPDATE

To understand the impact of updating the adapter for a single language on the merged model, we retrain the adapter of a specific language using additional data. We use sentiment analysis as a case study for this experiment. For this task, since English had the lowest performance among all the languages, we retrain the English adapter with an additional 5,000 examples. Merging the updated English adapter with the adapters for the other four languages showed an improved F1 score on English compared to the best merged model. We further observe that updating the English adapter not only improved the performance of the merged model for English, but we also see a performance improvement in three other languages, as seen in Table 2. The results suggest that updating a single language adapter can improve the performance not only on the updated language but also on other languages under consideration; however despite the improvements we were not able to surpass the baseline performance for sentiment analysis.

## 5.2 IMPACT OF MODEL SIZE

With model merging, the Llama-8b model achieves on-par results with the baselines. We investigate the impact of merging with a smaller LLMs via tha Llama-3b model. Similar to the Llama-8b experiments, we train language-specific models for sentiment analysis and summarization. For each task, we merge the language-specific models using the best hyperparameters and merging methods from the initial experiments. The Llama-3b model showed similar behavior as the Llama-8b model on merging, as seen in Figure 4. For summarization, the merged Llama-3b model achieved BertScore on par with the combined Llama-3b model trained on all languages together, and for sentiment analysis, the F1 score of the Llama-3b merged model is slightly lower than the combined Llama-3b

model. For both these tasks, this pattern is consistent with what we observed with the Llama-8b model. The Llama-3b model is slightly worse when compared to the Llama-8b model, which is expected given that the Llama-3b model has significantly lower number of parameters. Hence, this experiment indicates that model merging is size agnostic and can be applied to LLMs of different sizes, however, the absolute performance may vary depending on how small or large the LLM is.

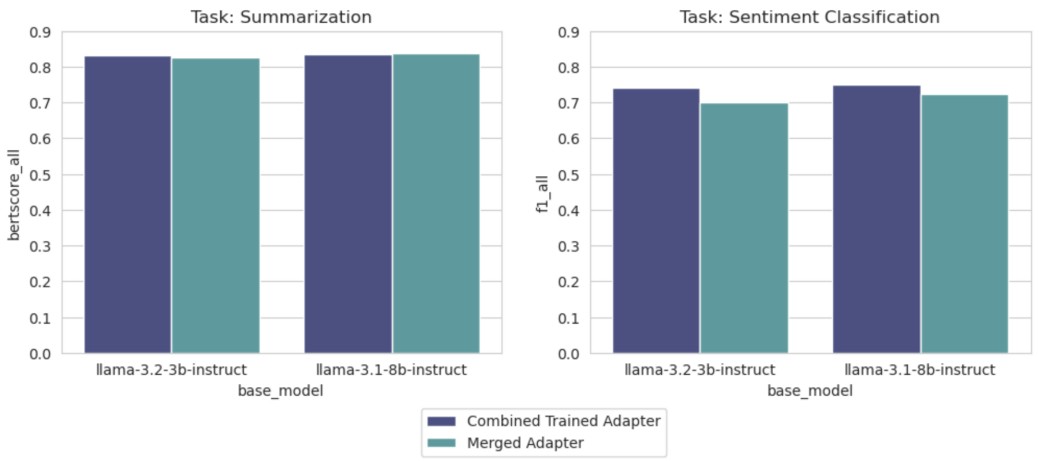

Figure 4: Performance comparison across model sizes

### 5.3 LANGUAGE CLUSTER-BASED MERGING

To understand if merging the models based on shared language properties improves the task performance, we cluster the languages based on their shared vocabulary. Specifically we group them in two clusters: European languages, namely English, German, and French, and East Asian Languages, Japanese and Chinese. As seen in Table 2, for sentiment analysis, we observe a decrease in performance for German and French compared to the best merged model, while the performance was on par for the other three languages. For summarization we observed that the language cluster based merging achieved on-par performance to the best merged model. On the commonsense reasoning task, we see an increase in accuracy for German, French, and Chinese, while for there was a slight decrease in accuracy for English. For Japanese however, we saw that the accuracy decreased by 3.4%. We can attribute this performance difference to the knowledge transfer during merging. When merging all languages, the merged models may inherit features from all the languages, while this transfer is limited with fewer languages. Moreover, the observations vary across tasks, indicating that merging the models based on language clusters may influence the performance differently based on the task under consideration. Overall, we did not observe a significant improvement with the language cluster-based model merging.

### 5.4 MULTITASK-MULTILINGUAL MERGING

Previous works on model merging show that merging task-specific models overall improves the performance on all tasks. We therefore investigate how multilingual-multitask merging impact the performance across different tasks. To this end we merge language-specific and task-specific models. We consider two scenarios: merging all language-specific models across all tasks together, and first merging language-specific models for a task, followed by merging across tasks. In both these scenarios, the overall performance degrades for all the tasks, as shown in Table 3. The performance decrease is between 2-5% for summarization and commonsense reasoning while for sentiment analysis, the F1 score drops by more than 5%. Comparing the two scenarios, merging all the language models together performs best for summarization and reasoning, while for sentiment analysis, merging language-specific models followed by task-specific merging works best. While the results generally indicate lower performance, hyperparameter tuning for task-specific and language-specific merging can improve the overall performance, and we leave this for future exploration.

Table 3: Multitask Multilingual Merging, $TIES_{All}$ refers to merging all language and task adapters together; $TIES_{Each}$ refers to merging language adapters for each task creating a single task adapter followed by merging independent task adapters together

| Model | Summarization (BertScore) | Reasoning (Accuracy) | Sentiment (F1) |
|---|---|---|---|
| $BEST_{Merged}$ | 0.836 | 0.800 | 0.724 |
| $TIES_{All}$ | 0.812 | 0.763 | 0.600 |
| $TIES_{Each}$ | 0.810 | 0.755 | 0.665 |

# 6 CASE STUDY

To understand the effectiveness of this technique for enterprises, we undertake a case study using a proprietary dataset. This task is similar to summarization, where an LLM processes unstructured data to identify relevant themes and provide supporting examples extracted from the input. It is supported in five languages, English, Spanish, German, French, and Japanese. The primary metric used is Aggregated Hallucination Rate[1], which computes the proportion of the number of LLM-generated examples that are not in the input. Similar to prior experiments, we first fine-tune language-specific models and a single multilingual model. We use the individual and the multilingual models as the baselines. Llama-3.1-8b-Instruct is used as the base model.

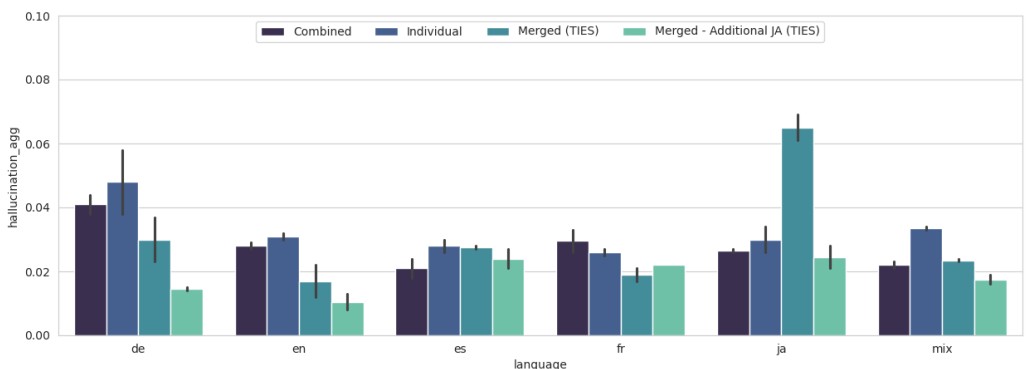

Figure 5: The aggregated hallucination rate across the languages (lower is better). The plot shows the scores for four models, two baselines, and the best performing merged model TIES. The scores for the model merged with updated Japanese data are also reported. The 'mix' language refers to having more than 1 language in the input that needs to be summarized

We merge the language-specific models using the three techniques described in Section 3.1. For this experiment, we assign differing weights to each language model based on some pre-defined condition. As seen in Figure 5, experimental results showed that the merged models achieved a comparable performance or improved the performance over the baselines for all languages except Japanese. We observed that Japanese had the highest hallucination rate among all the languages. Hence we retrained the Japanese model with more training data. Merging the retrained Japanese model with other language adapters not only improved the performance of the merged model on Japanese, but we also observed improved performance for other languages like English and German. This supports our initial observation from Section 5.1 that performance improvement may propagate across languages.

The experiment further demonstrates the effectiveness of language-specific model merging. As seen in Figure 5 and Figure 6 model merging allows us to save on training time and eventual training costs without compromising on the model performance. We were able to update the Japanese adapter at 37.5% of the cost via model merging as compared to the traditional method. Apart from computational efficiency, merging allows the hyperparameters for each language to be tuned separately depending on the business needs, giving more language-specific control.

---

[1]Lower hallucination rate is better.

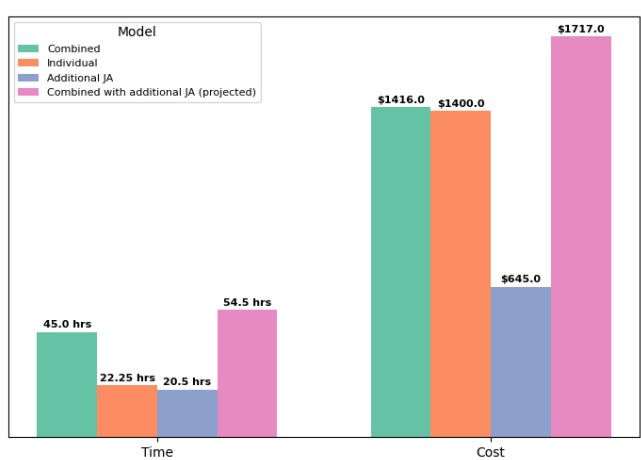

Figure 6: Training Speed and Cost Improvements

## 7 CONCLUSION

In this work we utilize existing language model merging techniques in a multilingual setting. Specifically, we use three techniques TIES, DARE, and KnOTS, and experiment on three tasks. Results indicate that TIES merging overall had the best performance across the three tasks. We further perform several ablations to evaluate the influence of merging based on language clusters, understand the impact of merging on model size, and identify the merits of merging across languages and models simultaneously. Through our experiments, we aimed to answer three research questions. We revisit and answer them here. For **RQ1**, our experiments showed that on two of the three tasks evaluated, multilingual model merging achieved comparable performance. While performing language-specific updates is more efficient with model merging compared to combined model retraining, it also improves the performance on languages other than the updated language, answering **RQ2**. Whereas for **RQ3**, we find that language cluster-based merging does not improve the performance over the model merged with all languages.

While the experiments show that multilingual model merging can be efficient in terms of training and achieve comparable performance to combined dataset training on most tasks, the results can be further improved through hyperparameter tuning and using other merging techniques. As a part of the future work, we plan to explore additional LLM sizes and families, as well as investigate ways to improve individual adapter weight selection, while further improving multitask-multilingual merging performance.

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

# A  APPENDIX

## A.1  ADDITIONAL RESULTS

This section provides Table 4 showing the values of all the computed metrics for each of the merging techniques and hyperparameter combinations we experimented with, and Figure 7 that showcases the language wise model performance of the best merged model per task against the combined and individual baselines, highlighting the on par performance observed for Summarization and Reasoning and the exceptions with Sentiment Analysis.

## A.2  LANGUAGE VECTOR ORTHOGONALITY

Ilharco et al. (2023) shows that the improved performance of the merged model on different tasks can be attributed to lower interference among the merged task vectors. Hence to investigate if there is interference between the vectors for the different languages, we check the orthogonality among the language vectors for all three tasks. Language vector for a specific language is obtained by computing the difference between the weights of the fine-tuned model and the base model. In our

Table 4: Overall metric scores of all models for Summarization, Reasoning, and Sentiment Analysis

| Model | Summarization | | | | Reasoning | Sentiment | | |
|---|---|---|---|---|---|---|---|---|
| | ROUGE-1 | ROUGE-2 | ROUGE-L | BertScore | Accuracy | Precision | Recall | F1 |
| $L8b_{COMB}$ | 0.012 | 0.002 | 0.012 | 0.835 | 0.795 | 0.750 | 0.751 | 0.750 |
| $L8b_{INDV}$ | 0.011 | 0.001 | 0.010 | 0.828 | 0.793 | 0.691 | 0.670 | 0.671 |
| $L8b\_DTS_{d=1.0}$ | 0.012 | 0.001 | 0.012 | 0.836 | 0.794 | 0.731 | 0.727 | 0.723 |
| $L8b\_DTS_{d=0.5}$ | 0.010 | 0.001 | 0.010 | 0.830 | 0.800 | 0.720 | 0.717 | 0.712 |
| $L8b\_DT_{d=1.0}$ | 0.007 | 0.001 | 0.007 | 0.805 | 0.786 | 0.654 | 0.646 | 0.646 |
| $L8b\_DT_{d=0.5}$ | 0.007 | 0.001 | 0.007 | 0.798 | 0.768 | 0.673 | 0.670 | 0.666 |
| $L8b\_TS_{d=1.0}$ | 0.012 | 0.001 | 0.012 | 0.836 | 0.794 | 0.731 | 0.727 | 0.723 |
| $L8b\_TS_{d=0.5}$ | 0.014 | 0.002 | 0.013 | 0.835 | 0.793 | 0.719 | 0.717 | 0.711 |
| $L8b\_T_{d=1.0}$ | 0.007 | 0.001 | 0.007 | 0.805 | 0.786 | 0.654 | 0.646 | 0.646 |
| $L8b\_T_{d=0.5}$ | 0.011 | 0.001 | 0.010 | 0.830 | 0.800 | 0.729 | 0.727 | 0.724 |

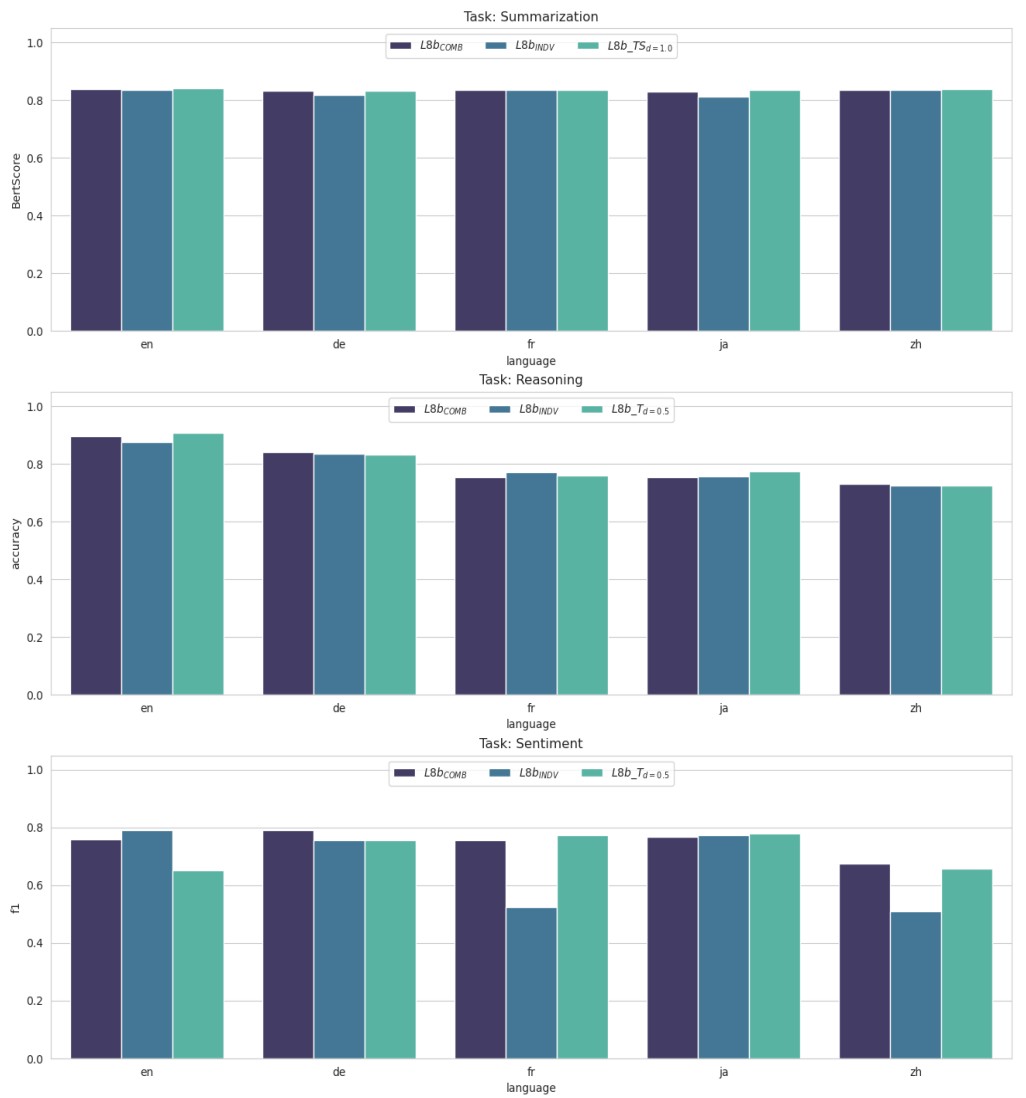

Figure 7: Language Wise Model Performance of the best merged model vs combined vs individually trained

case, since we use LoRA, that do not directly update the base model weights, we consider the

product of the weight matrics A and B obtained after fine-tuning as the language vector for the specific language. The cosine similarity between any two language vectors is computed to check the orthogonality between them. The similarity matrices are shown in Figure 8.

We hypothesized that since all the languages are trained on the same task, the language vectors would be similar and hence they may not be orthogonal to each other. However, the cosine similarity computations revealed that the language vectors for any two languages have comparatively lower similarity, especially for related languages like English and German. This indicates that although all languages learn the same task, they may have different semantic learning spaces to adapt to a specific task. Another possible factor in the lower similarity between the language vectors can be the amount of pre-training data used per language. While the Llama-3 pretraining data contained a significant amount of English data, the data for other languages was minimal. Hence, to adapt to a specific task, during fine-tuning, the weight updates required for English compared to other languages are smaller. Other factors like language-specific semantics, syntactic structures, as well as model tokenization, can also influence the similarity between the vectors. For sentiment analysis and reasoning, the similarities are comparatively lower than those for summarization, indicating the task influence.

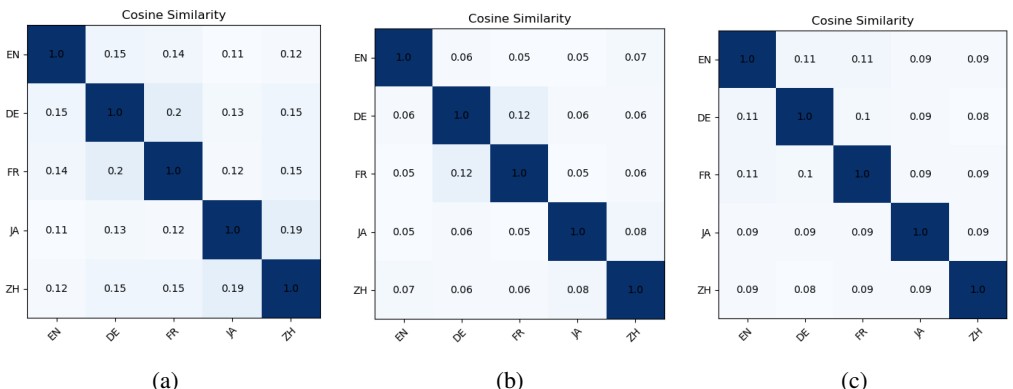

(a)          (b)          (c)

Figure 8: Cosine similarity between language vectors for each task. The similarity score is computed between each language pair

## A.3 LLM USAGE

LLMs were only used as writing assistants to help with grammar correction and rephrasing a few passages in this paper.

