# OpenReview forum: "Improving Training Efficiency via Language Specific Model Merging"
_ICLR.cc/2026/Conference — ICLR 2026 Conference Withdrawn Submission_

### Official Review · Reviewer_egJU · 2025-10-26

**Soundness:** 2
**Presentation:** 1
**Contribution:** 2
**Rating:** 2
**Confidence:** 4

**Summary:**

This paper investigates three existing model merging techniques (TIES, DARE, and KnOTS) for continued training/model merging across languages. The authors evaluate these methods in terms of performance and training efficiency on three tasks: Sentiment Analysis, Commonsense Reasoning, and Summarization. The main finding is that observations previously made for multi-task model merging (that it is more efficient and achieves competitive performance compared to joint training) also extend to multilingual settings.

**Strengths:**

* Continual pretraining for multilinguality is an under-studied area, and it is valuable to see a study analyzing this direction.
* The work provides confirmation that existing model merging techniques can be applied not only for task adaptation but also for language adaptation.

**Weaknesses:**

* The contributions are limited in scope. While not every paper needs to introduce a novel method, many aspects of the analysis are insufficiently developed..
* The problem of modular continual learning in multilingual settings has been explored in prior work (see [1]), and thus the claimed analytical contribution is not entirely novel. The method explored in [1] is not studied in this work.
* The discussion in the first few paragraphs of Section 4 is mostly speculative and does not provide concrete insights.
* The paper does not include experiments in a continual training scenario where K languages are present and a (K+1)th language is added **by simple continued training on the (K+1)th language (instead of model merging)**. This setup would better support the paper’s central claim that model merging is better.
* The paper is poorly written and hard to follow. Several experimental details are underspecified, and the exposition lacks clarity. Substantial revision would be required for the paper to meet publication standards.

[1] Blevins et al. Breaking the Curse of Multilinguality with Cross-lingual Expert Language Models. EMNLP 2024.

**Questions:**

* Figure 1: What is meant by an “inference model”? This is not clearly defined.
* Line 305: The statement “Retrain the adapter of a specific language using additional data” is ambiguous. What constitutes the additional data? What is the motivation for this experiment, and what does “retraining the adapter” mean?
* Section 6: The description of the task is too vague. Simply stating that the “tasks look like summarization” is insufficient without specifying the nature of the inputs and outputs.
* Line 190: The selection and usage of hyperparameters are not described in sufficient detail. It is unclear how these hyperparameters are used in these methods, or differ across methods.

---

### Official Review · Reviewer_Usag · 2025-10-29

**Soundness:** 2
**Presentation:** 3
**Contribution:** 2
**Rating:** 4
**Confidence:** 3

**Summary:**

This paper studies language model merging techniques in a multilingual setting. The authors empirically examine three existing methods—TIES, DARE, and KnOTS—and evaluate them on downstream tasks including summarization, commonsense reasoning, and sentiment analysis.

**Strengths:**

* The paper is well-structured and easy to follow.
* The authors conduct several interesting analyses regarding cross-lingual transfer and the impact of language family.
* The authors provide a case study, demonstrating its practical usage.

**Weaknesses:**

The authors aim to examine existing methods in a multilingual setting. I was hoping they would provide some practical recommendations based on their results. However, I am afraid it is difficult to draw firm conclusions based on their results.

I don't think the authors really examaine three merging techniques, as they claim. There is only one merging techniquee, TIES, used in all variants. The other two DARE, KnOTS are used to together with TIES.

**Questions:**

N/A

---

### Official Review · Reviewer_hSjm · 2025-10-30

**Soundness:** 2
**Presentation:** 3
**Contribution:** 2
**Rating:** 2
**Confidence:** 4

**Summary:**

This paper investigates merging of multilingual models. The study applies three model merging techniques (TIES, DARE, KnOTS) on three tasks (sentiment analysis, abstractive summarization, and commonsense reasoning). The authors find that on 2 out of 3 tasks, merging can achieve comparable performance to fine-tuning on the combined data, at lower costs. They perform additional ablations including on a proprietary dataset where they find that language cluster-based merging does not improve performance over merging with all languages.

**Strengths:**

1. Given the cost of running and fine-tuning large, the paper’s topic of investigating ways to improve models more efficiently is timely.
2. The paper studies three relatively recent model merging methods.
3. The paper conducts additional ablations that shed more light on the results.

**Weaknesses:**

1. The study only applies merging to models fine-tuned with LoRA. Given that LoRA fine-tunes embeddings in a low-dimensional space, it’s unclear what impact this has on the space of merged models. It would be important to evaluate merging of models fine-tuned without LoRA to understand whether the results generalize to the standard setting.

2. The study only uses a single base model (Llama 3.1-8B-Instruct; Llama-3B for the ablation) that by now is somewhat outdated so it’s unclear whether similar results are observed with more performant recent multilingual models (e.g., Qwen3, CommandR7B, etc). Given the use of LoRA, which makes fine-tuning more efficient, I also would have liked to see experiments with a much larger model, which would have made the findings more relevant for current practitioners.

3. Simple baselines such as merging with a simple or weighted averaging are missing from the evaluation so it’s unclear whether the particular algorithmic choices make a difference or simple merging is sufficient.

4. The novelty of the work seems to be limited. Prior work ([https://arxiv.org/abs/2410.10801v1](https://arxiv.org/abs/2410.10801v1)) already evaluates TIES, DARE, and other merging methods in a multilingual setting (also comparing them to training data mixing) and investigates merging of monolingually fine-tuned models so it’s unclear how this study meaningfully goes beyond prior work.

5. The datasets used are not very common or standard these days. The only reference related to the sentiment analysis dataset points to a HuggingFace repo; I wasn’t able to find a technical report or more information regarding how the data was created and its quality. Sentiment analysis and abstractive summarization as tasks are also somewhat outdated. I would have liked to see results on some more contemporary multilingual tasks including IF (mAlpaca, mDolly, etc).

6. The results across different benchmarks are very similar. It would be important to do statistical significance testing to understand which gains are actual improvements and which are just noise.

**Questions:**

1. Can you provide further context on why you used these three datasets for evaluation in your paper?

---

### Official Review · Reviewer_E4As · 2025-11-02

**Soundness:** 3
**Presentation:** 1
**Contribution:** 2
**Rating:** 2
**Confidence:** 4

**Summary:**

This paper studies the application of existing model merging methods (TIES, DARE, KnOTS) to a multilingual single-task setting. The paper compares the merging approches against two baselines, including training individual models for each language and training a single model on a cmobined multilingual dataset. Experimental results on three tasks, i.e., sentiment analysis, abstractive summarization, and commonsense reasoning, show that model merging methods can achieve comparable results with combined-dataset baseline but requiring lower training cost.

**Strengths:**

- **Practical and Relevant Research Direction**:  The paper addresses a practical and highly relevant challenge in deploying large language models: the efficient creation and maintenance of models that support multiple languages. This research is important for extending the capabilities of LLMs beyond high-resource languages. The authors' choice to explore model merging for this purpose is well-motivated; given the promising results of model merging in multi-task learning, its application to the multilingual setting is a natural and worthwhile line of inquiry.
- **Analysis of Cost**: The paper goes beyond standard accuracy performan to provide an analysis of training cost, highlighting the advantage of model merging. The comparisons in Figures 3 and 6 effectively underscore the primary practical advantage of the proposed merging workflow, particularly in scenarios involving the addition or updating of language support. This provides a clear, quantitative argument for the method's utility from an operational perspective.

**Weaknesses:**

- **Unclear Motivation and Novelty**: The central claim that "multilingual model merging is underexplored" is not entirely accurate. The paper itself cites recent works like AdaMergeX and others that leverage model merging in a multilingual or cross-lingual context such as MAD-X. The attempt to distinguish this work by stating it focuses on "effectiveness" while others focus on "transfer learning" is confusing. Model merging is fundamentally a mechanism for cross-lingual transfer, so this distinction feels artificial.
- **Insufficient Experimental Baselines**: The experimental setup is not solid enough for a comprehensive exploration of multilingual model performance. A significant omission is the lack of comparison to strong, standard cross-lingual transfer baselines or even a simple but often effective approach that use machine translation as a pipeline (translate-train / translate-test). Without comparing to such methods, it is difficult to assess the true effectiveness of the model merging approach in the broader context of cross-lingual learning.
- **Marginal and Inconsistent Performance Gains** The experimental results do not present a strong case for the superiority of model merging in terms of performance. In many cases, the merged models perform on par with, and sometimes worse than, the simple COMB baseline (training on the combined dataset). For example, in Table 1, the performance differences between the best-merged model and the COMB baseline are marginal for summarization and reasoning.
- **Low-Quality Presentation and Figures**: The paper's quality of presentation does not meet the standards for ICLR. (1) Uninformative Figures: Figures 2 and 4 are composed of bar charts where most bars have very similar heights, making it difficult to visually discern the small performance differences being discussed. This format fails to effectively convey the key results. (2) Formatting Issues: There are numerous small but distracting formatting errors. The font in Figure 3 is too small to be legible. The font in Figure 4 appears blurred. Captions for Figures 1, 4, and 5 are missing periods. There is inconsistent capitalization in the captions / labels of Figure 5 and Figure 6. These issues suggest a lack of careful proofreading.

**Questions:**

- The paper positions its contribution by differentiating from prior works like AdaMergeX, which it claims use "transfer learning." Could you please clarify this distinction? Since model merging is inherently a technique for transferring knowledge between models (which store knowledge in different languages), how is your investigation fundamentally different from studying cross-lingual transfer via model merging?

Why were other well-established cross-lingual transfer learning baselines or machine-translation pipeline baselines, not included in your experiments? Comparing against such methods would provide a more complete picture of where language-specific model merging stands in the landscape of multilingual solutions.

---

### Note · Authors · 2025-11-17

I have read and agree with the venue's withdrawal policy on behalf of myself and my co-authors.